# Analysis of Lipids in Green Coffee by Ultra-Performance Liquid Chromatography–Time-of-Flight Tandem Mass Spectrometry

**DOI:** 10.3390/molecules27165271

**Published:** 2022-08-18

**Authors:** Yijun Liu, Min Chen, Yimin Li, Xingqin Feng, Yunlan Chen, Lijing Lin

**Affiliations:** 1Hainan Key Laboratory of Storage & Processing of Fruits and Vegetables, Agricultural Products Processing Research Institute, Chinese Academy of Tropical Agricultural Sciences, Zhanjiang 524001, China; 2Key Laboratory of Tropical Crop Products Processing of Ministry of Agriculture and Rural Affairs, Zhanjiang 524001, China; 3College of Tropical Crops Institute, Yunnan Agricultural University, Pu’er 665099, China

**Keywords:** green coffee, ultra-performance liquid chromatography–time-of-flight tandem mass spectrometry, lipid

## Abstract

Lipid components in green coffee were clarified to provide essential data support for green coffee processing. The types, components, and relative contents of lipids in green coffee were first analyzed by ultra-performance liquid chromatography–time-of-flight tandem mass spectrometry (UPLC-TOF-MS/MS). The results showed that the main fatty acids in green coffee were linoleic acid (43.39%), palmitic acid (36.57%), oleic acid (8.22%), and stearic acid (7.37%). Proportionally, the ratio of saturated fatty acids/unsaturated fatty acids/polyunsaturated fatty acids was close to 5.5:1:5.2. A total of 214 lipids were identified, including 15 sterols, 39 sphingosines, 12 free fatty acids, 127 glycerides, and 21 phospholipids. The main components of sterols, sphingosines, free fatty acids, glycerides, and phospholipids were acylhexosyl sitosterol, ceramide esterified omega-hydroxy fatty acid sphingosine, linoleic acid, and triglyceride, respectively. UPLC-TOF-MS/MS furnished high-quality and accurate information on TOF MS and TOF MS/MS spectra, providing a reliable analytical technology platform for analyzing lipid components in green coffee.

## 1. Introduction

Coffee, mainly composed of protein, fat, total sugar, crude fiber, water, caffeine, water leachate, and free amino acids, is a genus of coffee in the Rubiaceae family. Taking Yunnan coffee as an example, the protein content is 14.0–17.7%, caffeine 1.02–1.33%, hydrolysable sugar 9.4–11.4%, acid lytic sugars 31.4–40.4%, crude fiber 21.50–28.74%, free amino acids 0.7–1.2%, and fat 4.7–7.1% [1]. Functional components in green coffee, mainly composed of alkaloids, phenolic acids, flavonoids, and terpenoids, play an important role in contributing to biological functions such as lowering blood sugar and protecting the liver and nerves [2].

Researchers have focused on coffee pretreatment processes, roasting methods, and coffee types [3,4,5,6,7]. For example, Yu et al. [8] used headspace solid-phase microextraction (HS-SPME) coupled with gas chromatography–mass spectrometry (GC-MS) to identify 82, 72, and 76 volatile organic compounds (VOCs) from green coffee roasted at three roasting speeds (namely, fast roast, medium roast, and slow roast), respectively, and the different roasting speeds affected the types and contents of VOCs. Juerg et al. [9] investigated the effect of roasting temperature and time on VOCs in green coffee. They found significant differences in aroma kinetic properties between high- and low-temperature conditions, and the concentration of compounds such as pyridine and dimethyl trisulfide in the aroma declined sharply. Some compounds increased when the temperature exceeded a certain level.

The coffee flavor was used as a critical indicator to assess coffee quality [10], and the results showed that fat in green coffee played a crucial role in flavor [11]. However, there have been few studies on the lipid analysis of coffee oil. Lipids play an essential physiological function in plant and animal growth and are closely related to human metabolism. However, there was a diversity of lipid structures, with over 40,000 lipids in the existing LIPIDMAPS lipid database and a narrow mass distribution range (0–1000 Da) [12], which posed a great difficulty for our analytical work. With the development of analytical techniques such as ultra-performance liquid chromatography–mass spectrometry (UPLC-MS), the analyses of lipids in *Prinsepia utilis Royle* oil and other samples have significantly developed. Among them, ultra-performance liquid chromatography–time-of-flight tandem mass spectrometry (UPLC-TOF-MS/MS) has been used to detect lipid components in samples because of its high detection sensitivity, short analysis time, simple pretreatment, and separation of lipid components at the mass spectrometry ion source. Xie et al. [13] performed qualitative and quantitative analysis of twenty triglyceride (TAG) molecules in cold-pressed rapeseed oil obtained before and after microwave pretreatment using direct injection multiplexed neutral loss scanning tandem mass spectrometry, and the results showed that the method could be applied to the detection of large sample volumes. This study aimed to establish a lipid analysis method based on high-performance liquid chromatography–time-of-flight tandem mass spectrometry (UPLC-TOF-MS/MS) and apply it to the analysis of lipids in green coffee and provide basic data for the development and utilization of green coffee. Meanwhile, profiling the microscopic lipid composition in green coffee helped reveal its functional mechanism.

## 2. Results and Discussion

### 2.1. Analysis of Fatty Acid Composition in Green Coffee

The relative percentages were calculated according to the chromatographic peak area normalization method concerning the time characterization of each fatty acid standard. The content of green coffee was 111.48 ± 3.56 mg/g, and its fatty acid composition and relative percentages were palmitic acid (C16:0) 36.57%, stearic acid (C18:0) 7.37%, oleic acid (C18:1n9c) 8.22%, linoleic acid (C18:2n6c) 43.39%, linolenic acid (C18:3n3) 1.13%, arachidic acid (C20:0) 2.56%, gadoleic acid (C20:1) 0.26%, behenic acid (C22:1) 0.26%, and behenic acid (C22:0) 0.50%. The fatty acids of green coffee were mainly composed of palmitic and linoleic acids, both of which were above 35%, followed by oleic and stearic acids. Koshima et al. [14] determined the fatty acid composition in green coffee oil using gas chromatography, and the results were consistent with the present experiment, except for with behenic acid. In addition, the variability in the types of coffee led to differences in the fatty acid types and contents in the results of this study and the analysis of Hong et al. [15].

According to their saturation, fatty acids are divided into saturated fatty acids (SFAs), monounsaturated fatty acids (MUFAs), and polyunsaturated fatty acids (PUFAs), which have different nutritional values. Green coffee contained 47% saturated fatty acids and 53% unsaturated fatty acids, of which 8.48% were monounsaturated fatty acids and 44.52% were polyunsaturated fatty acids, and the ratio of fatty acid composition (SFA/MUFA/PUFA) was approximately 5.5:1:5.2.

### 2.2. Identification of Lipids

In lipid molecules in mass spectrometry, relatively weak chemical bonds in the molecule are broken due to ionization, forming specific product ions or neutral lost fragment ions. This study identified sterols, sphingosines, glycerolipids, phospholipids, and fatty acids in green coffee from the perspective of mass spectrometry cleavage patterns. The fatty acids were identified as ASG (acylhexose glutathione) 29:1; O; Hex; FA 16:0 in acylhexosyl sitosterol (AHexSIS), ceramide esterified omega-hydroxy fatty acid-sphingosine (Cer_EOS) in Cer 60:12, and diacylglycerol (DG) in DG 34:2 |DG 16:0_18:2 as examples to analyze their mass spectrometric behaviors and fracture mechanisms in detail.

The molecular species of the compounds were identified by retention time, isotope distribution, MS mass-to-charge ratio, and MS/MS secondary mass spectrometry pattern in positive and negative ion modes. ASG had an excellent mass spectrometric response in both positive and negative ion modes. In positive ion mode, specific diagnostic fragment ions could be generated to identify its sterol lipid molecular species. In negative ion mode, fatty acid acyl chain composition could be identified by forming free state fatty acid fragment ions through ester bond breakage.

Figure 1A represents the MS/MS spectrum of ASG 29:1 Hex; FA 16:0 in positive ion mode; m/z 832.6495 was the precursor ion [M+NH4]^+^, and m/z 397.3795 was the diagnostic fragment ion of the acyl hexose glutamate ST 29:1+ [C29H49]^+^ sterol ester. Figure 1B displays the MS/MS mass spectra of ASG 29:1 Hex; FA 16:0 in the negative ion mode, where *m*/*z* 873.6855 was the precursor ion [M+CH3COO]^+^, and *m*/*z* 255.2280 was the characteristic fragment ion [FA 16:0-H]^−^. Since it was an ion formed by the loss of an H in the negative ion mode of the fatty acid formed in the free state after the ester bond was broken, it could be inferred that the fatty acid chain of this compound was Hex; FA 16:0. Figure 1C shows the MS/MS spectrum of DG 34:2 (16:0_18:2) in positive ion mode. From the figure, *m*/*z* 610.5323 could be tentatively determined as [M+NH4]^+^ of DG34:2, *m*/*z* 575.5026 represented [M+NH4-NH3-H2O]^+^, which was the fragment ion formed after the precursor ion [M+NH4]^+^ lost NH_3_ and H_2_O, and *m*/*z* 313.2735 and *m*/*z* 337.2728 represented [M+NH4-NH3-FA18:2]^+^ and [M+NH4-NH3-FA16:0]^+^, respectively, both of which were diagnostic fragment ions for fatty acid acyl chain characteristics. The monoglyceride sheet ions 16:0 DMAG+ and 18:2 DMAG+ formed after the loss of one fatty acid FA18:2 and FA16:0 from the precursor ion *m*/*z* 610.5323, respectively. Both di- and triglycerides were nonpolar lipids, forming ammonium addition ions [M+NH4]^+^ only in the positive ion mode, with no display in the negative ion mode, and the characteristic fragments were monoglyceride fragments and diglyceride fragments formed after the loss of one fatty acid, respectively. Figure 1D shows the MS/MS spectra of Cer 60:12;4O|Cer 42:9;3° (FA 18:2) in negative ion mode. *m*/*z* 910.7186 was the precursor ion [M-H]^−^ of Cer 60:12;4°, *m*/*z* 648.4904 was the fragment ion after the neutral loss of FA 18:2 of the precursor ion, and *m*/*z* 279.2293 was the characteristic fragment ion [FA 18:2-H]^−^.

### 2.3. Analysis of Lipid Composition in Green Coffee

UPLC-TOF-MS/MS analyzed the lipids in green coffee, and information on the precise relative molecular masses, isotopic distribution, and secondary mass spectrometry cleavage fragments of the lipids were obtained in compound scanning mode. As shown in Figure 2, a total of 214 lipids were identified in green coffee, including fifteen sterols, thirty-nine sphingomyelins, twelve free fatty acids, 127 glycerides, and twenty-one phospholipids. The above fifteen sterols mainly included four types of acylhexosyl campesterol (AHexCAS), five acylhexosyl sitosterols (AHexSIS), three acylhexosyl stigmasterols (AHexSTS), and three stigmasterol hexosides (SHex). The thirty-nine sphingosine species included seven types of ceramide alpha-hydroxy fatty acid phytosphingosine (Cer_AP), twenty-one types of ceramide esterified omega-hydroxy fatty acid dihydrosphingosine (Cer_EOS), one ceramide esterified omega-hydroxy fatty acid dihydrosphingosine (Cer_EODS), four ceramide nonhydroxy fatty acid phytosphingosine (Cer_NP), and six Hexosylceramide alpha-hydroxy fatty acid phytosphingosine (HexCer_AP). The twenty-one phospholipids included five types of phosphatidylcholine (PC), seven phosphatidylethanolamines (PE), one phosphatidylglycerol (PG), and eight phosphatidylinositols (PI). The 127 glycerol esters included twenty-four types of diacylglycerol (DG), three ether-linked triacylglycerols (EtherTG), three oxidized triglycerides (OxTG), one phosphatidylethanolamine (PE), one phosphatidylglycerol (PG), eight phosphatidylinositols (PI), twenty-five triglycerides (OxTG), seventy-one triglycerides (TG), and four monoacylglycerols (MG).

As shown in Table 1, the total number of carbon atoms in the fatty acid side chains of lipids in green coffee was 28–64, and the double bond number was 0–13. Most lipids contained at least one fatty acid side chain with a carbon number of 18 and a double bond number of 0–3. Among the sphingomyelinols, Cer_EOS had the highest number of double bonds. The number of carbon atoms of AHexCAS in sterols was 28, and the double bond number was one. The number of carbon atoms of AHexSTS and AHexSIS was 29, and the number of double bonds was one and two, respectively. The number of carbon atoms of Cer_EOS in sphingosine was 54–64, and the number of double bonds was 7–13. The number of carbon atoms of Cer_NP was 34–44, and the double bond number was 0–1. The number of carbon atoms of HexCer_AP was 36–44, and the number of double bonds was one. FA had a carbon atom number of 14–24 and a double bond number of 0–3. The number of carbon atoms of PC in phospholipids was 28, and the double bond number was 1–4. PE had a carbon atom number of 32–38 and a double bond number of 0–4. PG had a carbon atom number of 36 and a double bond number of zero. PI had a carbon atom number of 32–40 and a double bond number of 0–4. The number of carbon atoms of DG in glycerolipids was 32–4, and the number of double bonds was 0–5. EtherTG had a carbon atom number of 53–59 and a double bond number of 2–5. OxTG had a carbon atom number of 50–58 and a double bond number of 1–8. TG had a carbon atom number of 48–62 and a double bond number of 0–7. MG had a carbon atom number of 16–20 and a double bond number of 0–1.

### 2.4. Lipids’ Content in Green Coffee

Since the mass spectra of lipids of the same class under the same detection conditions should be similar and comparable, the peak areas of the extracted ion chromatographic peaks from the primary mass spectra in green coffee were used in this experiment for the quantitative calculation of similar lipids, as shown in Figure 3.

As shown in Figure 3, green coffee was mainly dominated by glycerides and fatty acids, 90.96 mg/g and 18.23 mg/g, respectively, followed by phospholipids 0.27 mg/g, sphingomyelin 0.076 mg/g, and sterols 0.016 mg/g, of which phospholipids were mainly PI and PC, with 0.188 mg/g and 0.066 mg/g, accounting for 70.38% and 24.70% of the total phospholipids, respectively. The sterols were mainly AHexSIS and SHex, with 0.007 mg/g and 0.005 mg/g, accounting for 44.68% and 33.64% of the total sterols, respectively. The sphingosine was mainly Cer_EOS at 0.056 mg/g, accounting for 73.69% of the total sphingosine. The glycerol esters were mainly TG with 74.632 mg/g, accounting for 82.04% of the total glycerol esters. Dietary triglycerides are the main component of vegetable oils, and their main functions are to supply and store energy, fix and protect internal organs, participate in the energy supply in several aspects of maternal and intrauterine fetal growth and development during pregnancy, and play a key role in lipid metabolism [16,17]. In addition to being absorbed by the body, gut microbes may also act upon dietary phospholipids to produce various phospholipids and choline. When the acetylcholine content in the brain increases, the speed of information transfer between nerve cells in the brain is accelerated and memory function is enhanced. In addition, intervention with phospholipid nutrients could improve the composition of arterial blood vessels, maintain esterase activity, improve the metabolism of lipids in the body, emulsify neutral esters and cholesterol deposited in the walls of blood vessels, promote the absorption of fats and fat-soluble vitamins, and improve intelligence and cellular activity [18,19], so green coffee is very rich in phospholipids and triglycerides and has a critical exploitation value.

Additionally, lipid composition and content vary significantly with raw materials, extraction processes, and other factors. Differential metabolites based on lipids can provide data support for food traceability [20,21], quality control during food processing, storage [22], etc. Wang et al. [23] used the phospholipid profiles of fish muscle to reveal the phospholipid oxidation and hydrolysis. Therefore, fish phospholipid molecules can be used as indicators of fish muscle freshness. Gao [24] used the UHPLC-MS method to screen 27 lipid molecules that could be used as biomarkers for identifying bacilli and fermented milk, providing a database for analyzing the effect of hot processing treatment on yogurt and fermented milk lipid quality. Liu et al. [25] demonstrated the efficiency of lipidomic analysis in identifying the geographic region and secretion period of goat milk in China. Similarly, this study’s results are instrumental in providing data support for the later identification of coffee species and the determination of used treatment processes.

## 3. Materials and Methods

### 3.1. Materials

Green coffee was provided by the Yunnan International Coffee Trading Center, made by wet processing technology, and originated from the variety of Catimor, which belongs to the Arabica coffee family. Triglyceride deuterium TAG 48:1 (15:0/18:1(D7)/15:0) and carbon XVII fatty acid methyl ester standard (internal standard) were purchased from Avanti Polar Lipids, pure chromatographic methanol, ammonia, chloroform, and hexane were purchased from Fisher, and chromatographic pure 10% ammonia was purchased from Shanghai Ampoule Experimental Technology Co. Ltd. (Shanghai, China). Chromatographic purity: dichloromethane for chromatographic purity was purchased from Sinopharm Chemical Reagent Co (Shanghai, China).

### 3.2. Methods

#### 3.2.1. Determination of Fatty Acid Composition by Gas Chromatography

The method was performed according to Wei et al. [26,27] with some modification of the parameters. Then, 1–2 mg of green coffee powder was added to the headspace vial, and 50 μL of 5 mg/mL of the internal standard carbon XVII fatty acid methyl ester, 2 mL of 5% concentrated sulfuric acid methanol solution, and 300 μL of toluene were pipetted sequentially. The headspace vial with an aluminum cap with a Teflon pad was sealed with a crimper, mixed with slight shaking, and extracted in a water bath at 95 °C for 1.5 h. At the end of extraction, the mixture was cooled to room temperature, 2 mL of 0.9% NaCl solution was added, mixed well, 1 mL of hexane was added for extraction, and the supernatant was centrifuged at 5000 rpm for 5 min in the supernatant bottle.

The GC-MS analytical conditions were equipped with a hydrogen flame ionization detector and DB-Fast FAME column (7890A gas chromatograph tandem hydrogen flame ionization detector, Agilent, Santa Clara, CA, USA). A total of 1.0 μL of the sample was driven through the column under nitrogen gas with an inlet temperature of 250 °C and a splitting ratio of 20:1, in which the initial temperature of the column was 80 °C for 5 min, 165 °C with a 40 °C/min for 1 min, 230 °C with a 4 °C/min for 6 min, and the detector temperature was 260 °C.

#### 3.2.2. Determination of Lipid Composition Using UPLC-TOF-MS/MS

The method was performed according to Xie et al. [28], with some modification of the parameters. Weigh approximately 20 mg of green coffee powder into a 10 mL tube, add 10 μL of 10 μg/mL of triglyceride deuterium internal standard and 2 mL of methanol, precipitate the protein overnight at −20 °C, add 2 mL of dichloromethane, vortex at 2000 rpm for 60 min and then add 2 mL of dichloromethane and 1.6 mL of ultrapure water, vortex and centrifuge, extract the lower clear, and add 4 mL of dichloromethane to extract the lower clear. The extraction was repeated twice, while the lower clear solution was collected three times. The supernatant was transferred into a 10 mL tube, blown dry with nitrogen, and then redissolved with 200 µL of dichloromethane/methanol (1:1, *v*/*v*), and the resulting solution was passed through a 0.22 μm organic filter membrane in the injection bottle for detection.

**UPLC–mass spectrometry conditions of the chromatographic system:** The analytical instrument was a Shimadzu UPLC LC-30A system (LC-30A liquid chromatograph, Shimadzu Corporation, Tokyo, Japan) equipped with a Phenomenex Kinete C18 column (100 × 2.1 mm, 2.6 µm). One microliter of the sample was pumped onto the column at a rate of 0.4 mL/min. The column temperature was 60 °C, and the sample chamber temperature was 4 °C. Gradient elution was performed using phase A (H_2_O:MeOH:ACN = 1:1:1, containing 5 mM NH_4_Ac) and phase B (isopropanol/acetonitrile = 5:1, containing 5 mM NH_4_Ac) with elution conditions of 20% B for 0.5 min, 40% B for 1.5 min, 60% B for 3 min, 98% B for 13 min, 20% B for 13 min, and 20% B for 17 min. In addition, the mass spectrometry system (Q-TOF-6600 Mass Spectrometer, AB Sciex, Concord, Ontario, Canada) was an AB Sciex TripleTOF^®^ 6600 coupled with an ESI source in positive and negative modes. The mass number collected by mass spectrometry ranged from *m*/*z* 100 to 1200, the ion spray voltage was 5500.00 V(+)/−4500 V(−), and the temperature was 600 °C.

### 3.3. Data Processing

Freely available MSDIAL, version 4.00 (http://prime.psc.riken.jp/Metabolomics_Software/MS-DIAL/index2.html, accessed on 5 November 2021), and commercially available software packages, Peak View, Master View, and Multiquanta (SCIEX, Washington, DC, USA), were used for lipid profiling. For lipid identification, the MS/MS spectrum of each feature was matched by MS-DIAL software with an integrated LipidBlast database [18]. Qualitative analysis of shotgun-MS data was performed using Lipid View software (v2.0, ABSciex, Concord, Ontario, Canada). Software parameter settings: Mass Tolerance = 0.5, Min % Intensity = 1, Minimum S/N = 10, Flow Injection Average Spectrum from Top = 30% TIC, Total Double Bonds ≤12.

### 3.4. Statistical Analysis

All data in this study were repeatedly measured three times, and data were analyzed statistically and significantly using IBM SPSS Statics analysis software and plotted using Origin Pro 2021.

## 4. Conclusions

In this study, 214 lipids were isolated and identified from green coffee by UPLC-TOF MS for the first time. The lipid content of green coffee lipids was 111.48 mg/g. The lipid components mainly consisted of sterols, sphingomyelin, free fatty acids, glycerides, and phospholipids at 0.016 mg/g, 0.076 mg/g, 18.23 mg/g, 90.96 mg/g, and 0.27 mg/g, respectively. The method combined high sensitivity, scanning speed, accuracy, and reproducibility. It processed the TOF MS/MS spectral information with high accuracy, providing a reliable analytical platform for the analysis of the lipid components of green coffee.

## Figures and Tables

**Figure 1 molecules-27-05271-f001:**
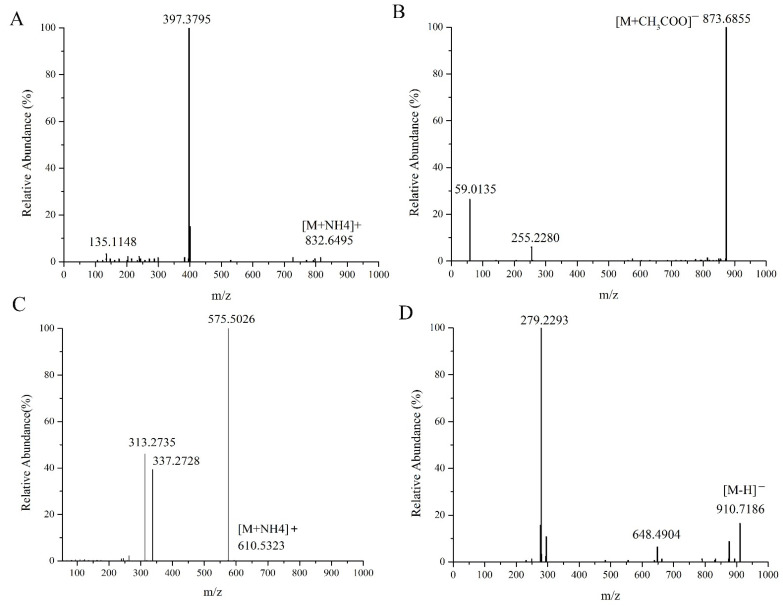
MS/MS spectra of ASG 29:1; O;Hex;FA 16:0 (**A**), DG 34:2 |DG 16:0_18:2 (**C**) in the positive ion mode and ASG 29:1;O;Hex;FA 16:0 (**B**), Cer 60:12 (**D**) in the negative ion mode.

**Figure 2 molecules-27-05271-f002:**
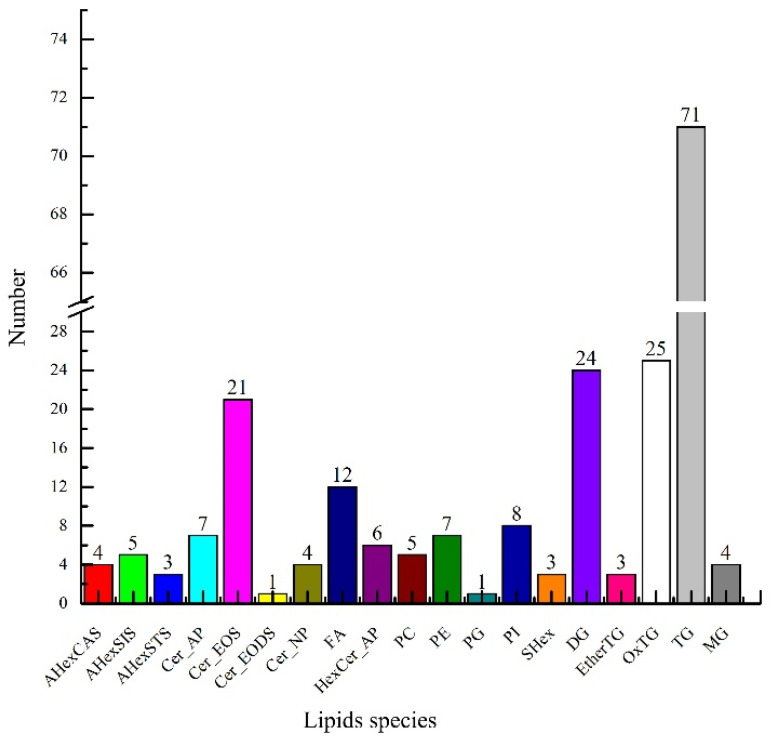
Lipid species in green coffee.

**Figure 3 molecules-27-05271-f003:**
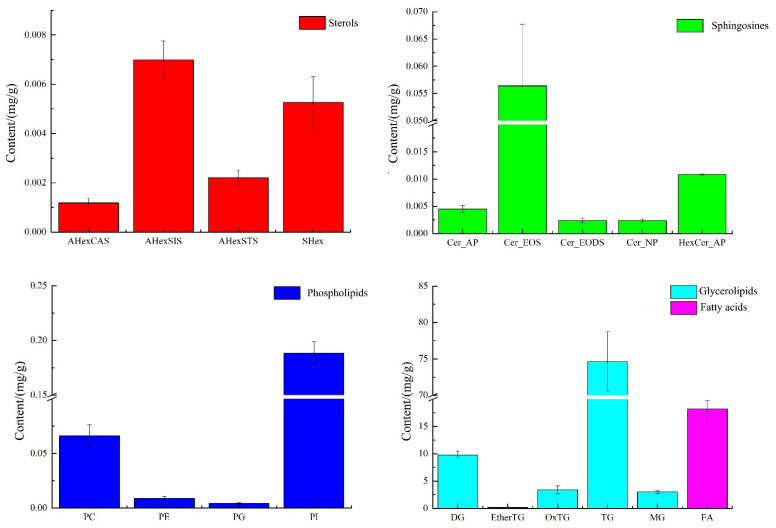
The composition of sterols, sphingosine, phospholipids, glycerides, and fatty acids in green coffee.

**Table 1 molecules-27-05271-t001:** Composition of the 214 lipids in green coffee.

NO	R.Time min	*m*/*z*	Lipid Name	Adduct Type	Formula	Ontology	Content μg/g
1	6.654	859.6649	ASG 28:1;O;Hex;FA 16:0	[M+CH3COO]−	C50H88O7	AHexCAS	0.61 ± 0.09
2	7.182	887.6971	ASG 28:1;O;Hex;FA 18:0	[M+CH3COO]−	C52H92O7	AHexCAS	0.20 ± 0.02
3	6.722	885.6755	ASG 28:1;O;Hex;FA 18:1	[M+CH3COO]−	C52H90O7	AHexCAS	0.09 ± 0.02
4	6.316	883.6581	ASG 28:1;O;Hex;FA 18:2	[M+CH3COO]−	C52H88O7	AHexCAS	0.29 ± 0.05
5	6.845	873.6815	ASG 29:1;O;Hex;FA 16:0	[M+CH3COO]−	C51H90O7	AHexSIS	2.79 ± 0.28
6	7.355	901.7115	ASG 29:1;O;Hex;FA 18:0	[M+CH3COO]−	C53H94O7	AHexSIS	0.97 ± 0.17
7	6.923	899.6962	ASG 29:1;O;Hex;FA 18:1	[M+CH3COO]−	C53H92O7	AHexSIS	0.56 ± 0.07
8	6.508	897.6824	ASG 29:1;O;Hex;FA 18:2	[M+CH3COO]−	C53H90O7	AHexSIS	2.03 ± 0.21
9	6.111	895.6605	ASG 29:1;O;Hex;FA 18:3	[M+CH3COO]−	C53H88O7	AHexSIS	0.64 ± 0.09
10	6.689	871.6605	ASG 29:2;O;Hex;FA 16:0	[M+CH3COO]−	C51H88O7	AHexSTS	1.01 ± 0.15
11	7.204	899.6945	ASG 29:2;O;Hex;FA 18:0	[M+CH3COO]−	C53H92O7	AHexSTS	0.56 ± 0.07
12	6.349	895.6656	ASG 29:2;O;Hex;FA 18:2	[M+CH3COO]−	C53H88O7	AHexSTS	0.64 ± 0.09
13	5.504	652.5873	Cer 40:1;4O|Cer 18:1;3O/22:0;(2OH)	[M-H]−	C40H79NO5	Cer_AP	0.51 ± 0.05
14	5.962	668.6175	Cer 41:0;4O|Cer 18:0;3O/23:0;(2OH)	[M-H]−	C41H83NO5	Cer_AP	0.18 ± 0.02
15	5.71	666.5975	Cer 41:1;4O|Cer 18:1;3O/23:0;(2OH)	[M-H]−	C41H81NO5	Cer_AP	0.22 ± 0.04
16	6.224	682.6332	Cer 42:0;4O|Cer 18:0;3O/24:0;(2OH)	[M-H]−	C42H85NO5	Cer_AP	1.60 ± 0.28
17	5.978	680.6147	Cer 42:1;4O|Cer 18:1;3O/24:0;(2OH)	[M-H]−	C42H83NO5	Cer_AP	1.42 ± 0.26
18	6.491	696.647	Cer 43:0;4O|Cer 18:0;3O/25:0;(2OH)	[M-H]−	C43H87NO5	Cer_AP	0.26 ± 0.03
19	6.241	694.6345	Cer 43:1;4O|Cer 18:1;3O/25:0;(2OH)	[M-H]−	C43H85NO5	Cer_AP	0.32 ± 0.08
20	9.529	838.7357	Cer 54:6;4O|Cer 38:5;3O(FA 16:0)	[M-H]−	C54H97NO5	Cer_EOS	0.11 ± 0.02
21	9.561	864.7435	Cer 56:7;4O|Cer 40:6;3O(FA 16:0)	[M-H]−	C56H99NO5	Cer_EOS	1.55 ± 0.42
22	9.186	862.7282	Cer 56:8;4O|Cer 21:1;2O/17:4;O(FA 18:2)	[M-H]−	C56H97NO5	Cer_EOS	13.16 ± 2.96
23	8.814	860.702	Cer 56:9;4O|Cer 40:8;3O(FA 16:0)	[M-H]−	C56H95NO5	Cer_EOS	0.26 ± 0.07
24	9.991	892.7689	Cer 58:7;4O|Cer 40:6;3O(FA 18:0)	[M-H]−	C58H103NO5	Cer_EOS	0.7 ± 0.16
25	9.637	890.7576	Cer 58:8;4O|Cer 40:7;3O(FA 18:0)	[M-H]−	C58H101NO5	Cer_EOS	5.81 ± 1.61
26	9.234	888.7432	Cer 58:9;4O|Cer 40:6;3O(FA 18:2)	[M-H]−	C58H99NO5	Cer_EOS	5.51 ± 0.94
27	8.846	886.7294	Cer 58:10;4O|Cer 40:7;3O(FA 18:2)	[M-H]−	C58H97NO5	Cer_EOS	14.84 ± 2.9
28	8.467	884.7048	Cer 58:11;4O|Cer 40:8;3O(FA 18:2)	[M-H]−	C58H95NO5	Cer_EOS	0.64 ± 0.13
29	10.396	920.797	Cer 60:7;4O|Cer 42:5;3O(FA 18:1)	[M-H]−	C60H107NO5	Cer_EOS	0.35 ± 0.04
30	10.062	918.7827	Cer 60:8;4O|Cer 42:5;3O(FA 18:2)	[M-H]−	C60H105NO5	Cer_EODS	2.40 ± 0.39
31	9.664	916.7686	Cer 60:9;4O|Cer 42:6;3O(FA 18:2)	[M-H]−	C60H103NO5	Cer_EOS	1.57 ± 0.38
32	9.299	914.7571	Cer 60:10;4O|Cer 42:7;3O(FA 18:2)	[M-H]−	C60H101NO5	Cer_EOS	3.34 ± 0.53
33	8.88	912.743	Cer 60:11;4O|Cer 42:8;3O(FA 18:2)	[M-H]−	C60H99NO5	Cer_EOS	2.20 ± 0.36
34	8.478	910.7285	Cer 60:12;4O|Cer 42:9;3O(FA 18:2)	[M-H]−	C60H97NO5	Cer_EOS	2.81 ± 0.54
35	8.111	908.7169	Cer 60:13;4O|Cer 42:9;3O(FA 18:3)	[M-H]−	C60H95NO5	Cer_EOS	0.27 ± 0.05
36	9.749	942.7889	Cer 62:10;4O|Cer 44:7;3O(FA 18:2)	[M-H]−	C62H105NO5	Cer_EOS	1.15 ± 0.29
37	9.346	940.7683	Cer 62:11;4O|Cer 44:8;3O(FA 18:2)	[M-H]−	C62H103NO5	Cer_EOS	0.20 ± 0.04
38	10.456	946.8187	Cer 62:8;4O|Cer 44:5;3O(FA 18:2)	[M-H]−	C62H109NO5	Cer_EOS	0.76 ± 0.16
39	10.089	944.8008	Cer 62:9;4O|Cer 44:7;3O(FA 18:1)	[M-H]−	C62H107NO5	Cer_EOS	0.54 ± 0.07
40	10.162	970.8248	Cer 64:10;4O|Cer 46:7;3O(FA 18:2)	[M-H]−	C64H109NO5	Cer_EOS	0.26 ± 0.06
41	10.835	974.8505	Cer 64:8;4O|Cer 40:7;3O(FA 24:0)	[M-H]−	C64H113NO5	Cer_EOS	0.36 ± 0.07
42	4.35	552.4949	Cer 34:1;3O|Cer 18:1;3O/16:0	[M-H]−	C34H67NO4	Cer_NP	0.98 ± 0.21
43	6.497	666.6363	Cer 42:0;3O|Cer 18:0;3O/24:0	[M-H]−	C42H85NO4	Cer_NP	0.73 ± 0.08
44	6.241	664.6186	Cer 42:1;3O|Cer 18:1;3O/24:0	[M-H]−	C42H83NO4	Cer_NP	0.38 ± 0.02
45	7.033	694.6718	Cer 44:0;3O|Cer 18:0;3O/26:0	[M-H]−	C44H89NO4	Cer_NP	0.29 ± 0.01
46	1.685	227.202	FA 14:0	[M-H]−	C14H28O2	FA	15.53 ± 1.00
47	2.334	255.2345	FA 16:0	[M-H]−	C16H32O2	FA	5727.91 ± 450.89
48	2.642	269.247	FA 17:0	[M-H]−	C17H34O2	FA	39.88 ± 1.53
49	2.92	283.2651	FA 18:0	[M-H]−	C18H36O2	FA	3086.39 ± 333.03
50	2.486	281.2492	FA 18:1	[M-H]−	C18H34O2	FA	1149.61 ± 143.83
51	2.058	279.2334	FA 18:2	[M-H]−	C18H32O2	FA	6403.60 ± 526.28
52	1.654	277.2187	FA 18:3	[M-H]−	C18H30O2	FA	94.98 ± 13.02
53	3.453	311.2959	FA 20:0	[M-H]−	C20H40O2	FA	854.03 ± 56.21
54	3.002	309.2787	FA 20:1	[M-H]−	C20H38O2	FA	58.09 ± 6.27
55	3.941	339.3272	FA 22:0	[M-H]−	C22H44O2	FA	315.63 ± 18.01
56	3.945	337.3143	FA 22:1	[M-H]−	C22H42O2	FA	0.38 ± 0.05
57	4.381	367.3573	FA 24:0	[M-H]−	C24H48O2	FA	483.68 ± 53.91
58	4.272	758.5736	HexCer 36:1;4O|HexCer 18:1;3O/18:0;(2OH)	[M-H]−	C42H81NO10	HexCer_AP	0.72 ± 0.10
59	4.605	786.608	HexCer 38:1;4O|HexCer 18:1;3O/20:0;(2OH)	[M-H]−	C44H85NO10	HexCer_AP	0.7 ± 0.06
60	5.02	814.6382	HexCer 40:1;4O|HexCer 18:1;3O/22:0;(2OH)	[M-H]−	C46H89NO10	HexCer_AP	5.47 ± 0.06
61	5.27	828.652	HexCer 41:1;4O|HexCer 18:1;3O/23:0;(2OH)	[M-H]−	C47H91NO10	HexCer_AP	0.25 ± 0.06
62	5.504	842.6694	HexCer 42:1;4O|HexCer 18:1;3O/24:0;(2OH)	[M-H]−	C48H93NO10	HexCer_AP	3.33 ± 0.12
63	5.973	870.6987	HexCer 44:1;4O|HexCer 18:1;3O/26:0;(2OH)	[M-H]−	C50H97NO10	HexCer_AP	0.36 ± 0.04
64	5.537	818.5863	PC 34:1|PC 16:0_18:1	[M+CH3COO]−	C42H82NO8P	PC	10.41 ± 1.93
65	5.067	816.5745	PC 34:2|PC 16:0_18:2	[M+CH3COO]−	C42H80NO8P	PC	30.21 ± 4.00
66	5.711	844.603	PC 36:2|PC 18:1_18:1	[M+CH3COO]−	C44H84NO8P	PC	5.25 ± 0.81
67	5.082	842.5946	PC 36:3|PC 18:1_18:2	[M+CH3COO]−	C44H82NO8P	PC	9.94 ± 1.88
68	4.687	840.5668	PC 36:4|PC 18:2_18:2	[M+CH3COO]−	C44H80NO8P	PC	10.24 ± 1.85
69	3.553	690.5078	PE 32:0|PE 16:0_16:0	[M-H]−	C37H74NO8P	PE	0.11 ± 0.01
70	4.989	716.5177	PE 34:1|PE 16:0_18:1	[M-H]−	C39H76NO8P	PE	0.33 ± 0.07
71	4.57	714.5034	PE 34:2|PE 16:0_18:2	[M-H]−	C39H74NO8P	PE	4.69 ± 1.06
72	5.036	742.5333	PE 36:2|PE 18:0_18:2	[M-H]−	C41H78NO8P	PE	1.17 ± 0.2
73	4.611	740.5236	PE 36:3|PE 18:1_18:2	[M-H]−	C41H76NO8P	PE	0.97 ± 0.19
74	4.335	738.5076	PE 36:4|PE 18:2_18:2	[M-H]−	C41H74NO8P	PE	1.55 ± 0.17
75	5.521	770.5684	PE 38:2|PE 20:0_18:2	[M-H]−	C43H82NO8P	PE	0.13 ± 0.05
76	4.335	777.5529	PG 36:0|PG 18:0_18:0	[M-H]−	C42H83O10P	PG	4.18 ± 0.68
77	3.716	809.5223	PI 32:0|PI 16:0_16:0	[M-H]−	C41H79O13P	PI	1.53 ± 0.14
78	3.76	835.5346	PI 34:1|PI 16:0_18:1	[M-H]−	C43H81O13P	PI	13.77 ± 1.51
79	3.55	833.5191	PI 34:2|PI 16:0_18:2	[M-H]−	C43H79O13P	PI	132.36 ± 7.38
80	3.351	831.4954	PI 34:3|PI 18:0_16:3	[M-H]−	C43H77O13P	PI	3.09 ± 0.41
81	3.847	861.5472	PI 36:2|PI 18:0_18:2	[M-H]−	C45H83O13P	PI	9.82 ± 1.10
82	3.588	859.5333	PI 36:3|PI 16:0_20:3	[M-H]−	C45H81O13P	PI	4.40 ± 0.38
83	3.375	857.5219	PI 36:4|PI 18:2_18:2	[M-H]−	C45H79O13P	PI	9.28 ± 0.84
84	4.54	919.6258	PI 40:1|PI 20:0_20:1	[M-H]−	C49H93O13P	PI	13.94 ± 2.16
85	3.613	621.4358	SG 28:1;O;Hex	[M+CH3COO]−	C34H58O6	SHex	0.49 ± 0.09
86	3.774	635.4494	SG 29:1;O;Hex	[M+CH3COO]−	C35H60O6	SHex	3.51 ± 0.76
87	3.655	633.4337	SG 29:2;O;Hex	[M+CH3COO]−	C35H58O6	SHex	1.27 ± 0.19
88	5.634	586.5302	DG 32:0|DG 16:0_16:0	[M+NH4]+	C35H68O5	DG	85.39 ± 13.32
89	5.225	584.5154	DG 32:1|DG 16:0_16:1	[M+NH4]+	C35H66O5	DG	2.35 ± 0.26
90	6.161	614.5614	DG 34:0|DG 16:0_18:0	[M+NH4]+	C37H72O5	DG	41.68 ± 4.04
91	5.701	612.5468	DG 34:1|DG 16:0_18:1	[M+NH4]+	C37H70O5	DG	206.95 ± 32.99
92	5.321	610.5339	DG 34:2|DG 16:0_18:2	[M+NH4]+	C37H68O5	DG	3804.73 ± 356.39
93	5.002	608.5155	DG 34:3|DG 16:0_18:3	[M+NH4]+	C37H66O5	DG	103.33 ± 10.96
94	6.698	642.5931	DG 36:0|DG 18:0_18:0	[M+NH4]+	C39H76O5	DG	25.20 ± 1.70
95	6.228	640.5801	DG 36:1|DG 18:0_18:1	[M+NH4]+	C39H74O5	DG	36.32 ± 4.29
96	5.813	638.5634	DG 36:2|DG 18:0_18:2	[M+NH4]+	C39H72O5	DG	535.91 ± 43.92
97	5.378	636.5472	DG 36:3|DG 18:1_18:2	[M+NH4]+	C39H70O5	DG	842.02 ± 76.18
98	5.022	634.5318	DG 36:4|DG 18:2_18:2	[M+NH4]+	C39H68O5	DG	3737.90 ± 263.65
99	4.723	632.5156	DG 36:5|DG 18:2_18:3	[M+NH4]+	C39H66O5	DG	107.66 ± 14.00
100	7.235	670.6293	DG 38:0|DG 16:0_22:0	[M+NH4]+	C41H80O5	DG	5.84 ± 0.24
101	6.76	668.6097	DG 38:1|DG 20:0_18:1	[M+NH4]+	C41H78O5	DG	11.47 ± 0.58
102	6.345	666.598	DG 38:2|DG 20:0_18:2	[M+NH4]+	C41H76O5	DG	145.25 ± 8.6
103	5.852	664.5808	DG 38:3|DG 20:1_18:2	[M+NH4]+	C41H74O5	DG	19.69 ± 1.43
104	5.482	662.5598	DG 38:4|DG 18:2_20:2	[M+NH4]+	C41H72O5	DG	6.07 ± 1.20
105	7.771	698.6593	DG 40:0|DG 20:0_20:0	[M+NH4]+	C43H84O5	DG	8.74 ± 0.61
106	7.302	696.6409	DG 40:1|DG 22:0_18:1	[M+NH4]+	C43H82O5	DG	2.27 ± 0.18
107	6.889	694.6287	DG 40:2|DG 22:0_18:2	[M+NH4]+	C43H80O5	DG	20.65 ± 1.90
108	8.282	726.6904	DG 42:0|DG 20:0_22:0	[M+NH4]+	C45H88O5	DG	16.76 ± 1.30
109	7.429	722.6627	DG 42:2|DG 24:0_18:2	[M+NH4]+	C45H84O5	DG	9.06 ± 0.21
110	8.762	754.7205	DG 44:0|DG 22:0_22:0	[M+NH4]+	C47H92O5	DG	12.20 ± 1.17
111	7.967	750.6893	DG 44:2|DG 26:0_18:2	[M+NH4]+	C47H88O5	DG	2.38 ± 0.38
112	9.622	876.8317	TG O-53:2|TG O-19:2_16:0_18:0	[M+NH4]+	C56H106O5	EtherTG	66.61 ± 7.81
113	8.966	898.827	TG O-55:5|TG O-19:1_18:2_18:2	[M+NH4]+	C58H104O5	EtherTG	66.51 ± 3.34
114	10.136	956.9129	TG O-59:4|TG O-19:2_18:2_22:0	[M+NH4]+	C62H114O5	EtherTG	17.46 ± 2.97
115	8.246	866.7822	TG 50:1;1O|TG 16:0_16:0_18:1;1O	[M+NH4]+	C53H100O7	OxTG	22.96 ± 6.44
116	7.946	864.7678	TG 50:2;1O|TG 16:0_16:0_18:2;1O	[M+NH4]+	C53H98O7	OxTG	504.38 ± 128.28
117	7.576	862.7489	TG 50:3;1O|TG 16:0_16:0_18:3;1O	[M+NH4]+	C53H96O7	OxTG	92.53 ± 28.21
118	7.095	860.737	TG 50:4;1O|TG 16:0_18:2_16:2;1O	[M+NH4]+	C53H94O7	OxTG	4.16 ± 0.80
119	8.721	894.8135	TG 52:1;1O|TG 16:0_18:0_18:1;1O	[M+NH4]+	C55H104O7	OxTG	10.19 ± 1.89
120	8.429	892.7969	TG 52:2;1O|TG 16:0_18:0_18:2;1O	[M+NH4]+	C55H102O7	OxTG	180.15 ± 42.4
121	7.973	890.783	TG 52:3;1O|TG 16:0_18:1_18:2;1O	[M+NH4]+	C55H100O7	OxTG	334.53 ± 82.77
122	7.596	888.7674	TG 52:4;1O|TG 16:0_18:2_18:2;1O	[M+NH4]+	C55H98O7	OxTG	1043.62 ± 205.51
123	7.235	886.7534	TG 52:5;1O|TG 16:0_18:2_18:3;1O	[M+NH4]+	C55H96O7	OxTG	273.4 ± 76.16
124	6.871	884.7346	TG 52:6;1O|TG 16:0_18:3_18:3;1O	[M+NH4]+	C55H94O7	OxTG	5.11 ± 1.5
125	8.915	920.8307	TG 54:2;1O|TG 16:0_20:0_18:2;1O	[M+NH4]+	C57H106O7	OxTG	49.40 ± 6.42
126	8.488	918.8096	TG 54:3;1O|TG 18:0_18:1_18:2;1O	[M+NH4]+	C57H104O7	OxTG	71.38 ± 16.36
127	8.09	916.7963	TG 54:4;1O|TG 18:0_18:2_18:2;1O	[M+NH4]+	C57H102O7	OxTG	198.9 ± 44.27
128	7.651	914.785	TG 54:5;1O|TG 18:1_18:2_18:2;1O	[M+NH4]+	C57H100O7	OxTG	191.35 ± 38.67
129	7.246	912.7684	TG 54:6;1O|TG 18:2_18:2_18:2;1O	[M+NH4]+	C57H98O7	OxTG	237.19 ± 41.13
130	6.889	910.7452	TG 54:7;1O|TG 18:2_18:2_18:3;1O	[M+NH4]+	C57H96O7	OxTG	44.14 ± 9.42
131	6.505	908.7288	TG 54:8;1O|TG 18:2_18:3_18:3;1O	[M+NH4]+	C57H94O7	OxTG	2.05 ± 0.34
132	9.363	948.8574	TG 56:2;1O|TG 16:0_22:0_18:2;1O	[M+NH4]+	C59H110O7	OxTG	11.20 ± 2.53
133	8.948	946.8486	TG 56:3;1O|TG 20:0_18:1_18:2;1O	[M+NH4]+	C59H108O7	OxTG	19.55 ± 2.11
134	8.584	944.8259	TG 56:4;1O|TG 20:0_18:2_18:2;1O	[M+NH4]+	C59H106O7	OxTG	57.93 ± 10.49
135	8.228	942.8184	TG 56:5;1O|TG 20:0_18:2_18:3;1O	[M+NH4]+	C59H104O7	OxTG	15.9 ± 2.36
136	7.754	940.8	TG 56:6;1O|TG 20:1_18:2_18:3;1O	[M+NH4]+	C59H102O7	OxTG	1.88 ± 0.35
137	9.783	976.8939	TG 58:2;1O|TG 20:0_20:0_18:2;1O	[M+NH4]+	C61H114O7	OxTG	4.72 ± 0.58
138	9.429	974.8809	TG 58:3;1O|TG 22:0_19:2_17:1;1O	[M+NH4]+	C61H112O7	OxTG	5.04 ± 1.09
139	9.043	972.858	TG 58:4;1O|TG 22:0_18:2_18:2;1O	[M+NH4]+	C61H110O7	OxTG	9.83 ± 0.45
140	9.524	824.7681	TG 48:0|TG 16:0_16:0_16:0	[M+NH4]+	C51H98O6	TG	138.29 ± 18.75
141	9.099	822.7542	TG 48:1|TG 14:0_16:0_18:1/TG 16:0_16:0_16:1	[M+NH4]+	C51H96O6	TG	15.50 ± 1.73
142	8.713	820.7387	TG 48:2|TG 14:0_16:0_18:2	[M+NH4]+	C51H94O6	TG	97.33 ± 17.62
143	8.972	834.7506	TG 49:2|TG 15:0_16:0_18:2	[M+NH4]+	C52H96O6	TG	41.98 ± 2.67
144	8.634	832.7381	TG 49:3|TG 16:0_15:1_18:2	[M+NH4]+	C52H94O6	TG	36.08 ± 4.77
145	9.954	852.7994	TG 50:0|TG 16:0_16:0_18:0	[M+NH4]+	C53H102O6	TG	83.42 ± 13.03
146	9.553	850.7856	TG 50:1|TG 16:0_16:0_18:1	[M+NH4]+	C53H100O6	TG	2563.84 ± 321.36
147	9.17	848.7705	TG 50:2|TG 16:0_16:0_18:2	[M+NH4]+	C53H98O6	TG	15,433.4 ± 1243.63
148	8.797	846.7551	TG 50:3|TG 16:0_16:0_18:3	[M+NH4]+	C53H96O6	TG	393.98 ± 70.02
149	8.357	844.7397	TG 50:4|TG 14:0_18:2_18:2	[M+NH4]+	C53H94O6	TG	60.27 ± 11.04
150	9.793	864.799	TG 51:1|TG 16:0_17:0_18:1	[M+NH4]+	C54H102O6	TG	19.04 ± 2.01
151	9.424	862.7835	TG 51:2|TG 16:0_17:0_18:2	[M+NH4]+	C54H100O6	TG	128.18 ± 5.95
152	9.066	860.7667	TG 51:3|TG 16:0_17:1_18:2	[M+NH4]+	C54H98O6	TG	26.49 ± 2.43
153	8.619	858.7512	TG 51:4|TG 15:0_18:2_18:2	[M+NH4]+	C54H96O6	TG	51.37 ± 6.93
154	10.373	880.8313	TG 52:0|TG 16:0_18:0_18:0	[M+NH4]+	C55H106O6	TG	37.16 ± 6.55
155	9.984	878.8175	TG 52:1|TG 16:0_18:0_18:1	[M+NH4]+	C55H104O6	TG	1301.48 ± 206.72
156	9.633	876.8012	TG 52:2|TG 16:0_18:0_18:2	[M+NH4]+	C55H102O6	TG	8843.51 ± 854.57
157	9.227	874.7888	TG 52:3|TG 16:0_18:1_18:2	[M+NH4]+	C55H100O6	TG	5707.01 ± 503.57
158	8.83	872.7711	TG 52:4|TG 16:0_18:2_18:2	[M+NH4]+	C55H98O6	TG	16,132.45 ± 1269.98
159	8.46	870.7558	TG 52:5|TG 16:0_18:2_18:3	[M+NH4]+	C55H96O6	TG	1460.96 ± 308.25
160	8.09	868.7389	TG 52:6|TG 16:0_18:3_18:3	[M+NH4]+	C55H94O6	TG	26.70 ± 6.32
161	10.198	892.8311	TG 53:1|TG 17:0_18:0_18:1	[M+NH4]+	C56H106O6	TG	7.41 ± 0.98
162	9.855	890.8171	TG 53:2|TG 17:0_18:0_18:2	[M+NH4]+	C56H104O6	TG	59.35 ± 5.01
163	9.452	888.8031	TG 53:3|TG 17:0_18:1_18:2	[M+NH4]+	C56H102O6	TG	39.12 ± 1.76
164	9.071	886.7877	TG 53:4|TG 17:0_18:2_18:2	[M+NH4]+	C56H100O6	TG	56.52 ± 6.54
165	8.707	884.7674	TG 53:5|TG 17:0_18:2_18:3	[M+NH4]+	C56H98O6	TG	13.14 ± 3.22
166	10.75	908.8629	TG 54:0|TG 16:0_18:0_20:0	[M+NH4]+	C57H110O6	TG	11.35 ± 2.42
167	10.395	906.8478	TG 54:1|TG 16:0_20:0_18:1	[M+NH4]+	C57H108O6	TG	524.05 ± 89.67
168	10.056	904.8348	TG 54:2|TG 16:0_20:0_18:2	[M+NH4]+	C57H106O6	TG	4254.17 ± 523.25
169	9.666	902.8181	TG 54:3|TG 18:0_18:1_18:2	[M+NH4]+	C57H104O6	TG	1873.53 ± 213
170	9.311	900.803	TG 54:4|TG 18:0_18:2_18:2	[M+NH4]+	C57H102O6	TG	3713.62 ± 253.08
171	8.892	898.7906	TG 54:5|TG 18:1_18:2_18:2	[M+NH4]+	C57H100O6	TG	2189.29 ± 160.77
172	8.488	896.7725	TG 54:6|TG 18:2_18:2_18:2	[M+NH4]+	C57H98O6	TG	4025.69 ± 559.05
173	8.107	894.7582	TG 54:7|TG 18:2_18:2_18:3	[M+NH4]+	C57H96O6	TG	308.01 ± 56.35
174	10.584	920.8643	TG 55:1|TG 16:0_21:0_18:1	[M+NH4]+	C58H110O6	TG	5.27 ± 1.86
175	10.259	918.8485	TG 55:2|TG 16:0_21:0_18:2	[M+NH4]+	C58H108O6	TG	74.37 ± 12.78
176	11.117	936.8935	TG 56:0|TG 16:0_18:0_22:0	[M+NH4]+	C59H114O6	TG	6.13 ± 0.95
177	10.775	934.882	TG 56:1|TG 18:0_20:0_18:1	[M+NH4]+	C59H112O6	TG	103.09 ± 22.17
178	10.452	932.8689	TG 56:2|TG 18:0_20:0_18:2	[M+NH4]+	C59H110O6	TG	1225.38 ± 193.62
179	10.098	930.851	TG 56:3|TG 20:0_18:1_18:2	[M+NH4]+	C59H108O6	TG	593.99 ± 75.39
180	9.75	928.8354	TG 56:4|TG 20:0_18:2_18:2	[M+NH4]+	C59H106O6	TG	1382.45 ± 140.08
181	9.381	926.8179	TG 56:5|TG 20:0_18:2_18:3	[M+NH4]+	C59H104O6	TG	153.43 ± 14.49
182	8.958	924.7989	TG 56:6|TG 20:1_18:2_18:3	[M+NH4]+	C59H102O6	TG	18.45 ± 2.23
183	10.644	946.8822	TG 57:2|TG 16:0_23:0_18:2	[M+NH4]+	C60H112O6	TG	88.96 ± 21.60
184	10.29	944.8629	TG 57:3|TG 21:0_18:1_18:2	[M+NH4]+	C60H110O6	TG	11.52 ± 2.00
185	9.943	942.8513	TG 57:4|TG 21:0_18:2_18:2	[M+NH4]+	C60H108O6	TG	27.24 ± 1.39
186	11.445	964.9293	TG 58:0|TG 16:0_18:0_24:0	[M+NH4]+	C61H118O6	TG	1.85 ± 0.27
187	11.137	962.9146	TG 58:1|TG 16:0_24:0_18:1	[M+NH4]+	C61H116O6	TG	33.88 ± 4.51
188	10.83	960.8977	TG 58:2|TG 16:0_24:0_18:2	[M+NH4]+	C61H114O6	TG	410.34 ± 71.01
189	10.481	958.8819	TG 58:3|TG 22:0_18:1_18:2	[M+NH4]+	C61H112O6	TG	97.2 ± 17.00
190	10.155	956.8651	TG 58:4|TG 22:0_18:2_18:2	[M+NH4]+	C61H110O6	TG	245.39 ± 42.5
191	9.836	954.8525	TG 58:5|TG 22:0_18:2_18:3	[M+NH4]+	C61H108O6	TG	13.39 ± 2.75
192	11.304	976.9302	TG 59:1|TG 16:0_25:0_18:1	[M+NH4]+	C62H118O6	TG	4.3 ± 0.62
193	11.01	974.9142	TG 59:2|TG 16:0_25:0_18:2	[M+NH4]+	C62H116O6	TG	48.72 ± 7.73
194	10.674	972.8973	TG 59:3|TG 23:0_18:1_18:2	[M+NH4]+	C62H114O6	TG	12.08 ± 2.99
195	10.344	970.8828	TG 59:4|TG 23:0_18:2_18:2	[M+NH4]+	C62H112O6	TG	39.24 ± 8.01
196	11.765	992.9612	TG 60:0|TG 16:0_20:0_24:0	[M+NH4]+	C63H122O6	TG	1.15 ± 0.28
197	11.465	990.9445	TG 60:1|TG 16:0_26:0_18:1	[M+NH4]+	C63H120O6	TG	8.11 ± 0.65
198	11.184	988.9325	TG 60:2|TG 16:0_26:0_18:2	[M+NH4]+	C63H118O6	TG	86.84 ± 5.72
199	10.851	986.9174	TG 60:3|TG 24:0_18:1_18:2	[M+NH4]+	C63H116O6	TG	46.77 ± 6.62
200	10.536	984.8977	TG 60:4|TG 24:0_18:2_18:2	[M+NH4]+	C63H114O6	TG	134.64 ± 16.89
201	10.243	982.8888	TG 60:5|TG 24:0_18:2_18:3	[M+NH4]+	C63H112O6	TG	6.53 ± 1.10
202	11.626	1004.964	TG 61:1|TG 18:0_25:0_18:1	[M+NH4]+	C64H122O6	TG	0.80 ± 0.07
203	11.346	1002.947	TG 61:2|TG 18:0_25:0_18:2	[M+NH4]+	C64H120O6	TG	7.92 ± 1.08
204	11.023	1000.929	TG 61:3|TG 25:0_18:1_18:2	[M+NH4]+	C64H118O6	TG	6.45 ± 1.03
205	10.731	998.915	TG 61:4|TG 25:0_18:2_18:2	[M+NH4]+	C64H116O6	TG	20.13 ± 3.34
206	12.064	1020.994	TG 62:0|TG 20:0_20:0_22:0	[M+NH4]+	C65H126O6	TG	1.39 ± 0.42
207	11.786	1018.98	TG 62:1|TG 18:0_26:0_18:1	[M+NH4]+	C65H124O6	TG	0.99 ± 0.04
208	11.514	1016.964	TG 62:2|TG 18:0_26:0_18:2	[M+NH4]+	C65H122O6	TG	9.31 ± 0.50
209	11.204	1014.949	TG 62:3|TG 26:0_18:1_18:2	[M+NH4]+	C65H120O6	TG	9.49 ± 0.85
210	10.908	1012.931	TG 62:4|TG 26:0_18:2_18:2	[M+NH4]+	C65H118O6	TG	20.87 ± 0.93
211	2.59	331.2843	MG 16:0	[M+H]+	C19H38O4	MG	1199.53 ± 168.11
212	3.09	359.3156	MG 18:0	[M+H]+	C21H42O4	MG	1711.34 ± 63.3
213	2.69	357.2999	MG 18:1	[M+H]+	C21H40O4	MG	51.45 ± 4.19
214	3.56	387.3469	MG 20:0	[M+H]+	C23H46O4	MG	39.13 ± 2.38

## Data Availability

The data presented in this study are available on request from the corresponding author.

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
