# Peer review of "Analysis of Lipids in Green Coffee by Ultra-Performance Liquid Chromatography–Time-of-Flight Tandem Mass Spectrometry"

_molecules, 2022, doi:10.3390/molecules27165271_

Round 1
Reviewer 1 Report
I have checked the revised version of the manuscript " Analysis of lipid of green coffee by ultra-performance liquidchromatography-time of flight-tandem mass spectrometry"
and the point-by-point response corrections fully answer the raised questions. Now, I recommend the publication of the revised version of the above manuscript on Molecules.
Author Response
no
Reviewer 2 Report
The authors have made significant improvements to the manuscript based on the reviewers' comments and the manuscript is suitable for publication. However, there are minor flaws that need to be corrected before it is accepted for publication The authors can try to again rectify the overall grammatical errors throughout the manuscript. I have listed some of the major errors below.
1. The terms 'green coffee' and 'green coffee oil' are used interchangeably throughout the text. Please keep it uniform
2. The author names are all capitalized in the manuscript and in references. Please use a uniform style and remove capitalization.
3. Section 2.1.1. "Determination of fatty acid composition by gas chromatography" is a title and should not be referenced. Remove the references for the title. The same applies to Section 2.1.2.
4. Section 2.2 is missing and make the title font and italics uniform.
5. Section 2.1.2. It is not clear what the authors mean by the statement "And the mass number range 120 for mass spectrometry was m/z 100-1200". Correct the grammatical errors throughout the manuscript.
6. Section 3.2: Please edit the statement ' It could be seen the Figure 1(A)'. Just mention as 'Fig 1A represents....'
7. Fig 3 legend should be 'The composition of sterols, sphingosine.."
Author Response
Responses to the Reviewers (molecules-1808702)
Dear Editor and reviewers,
Thank you for your comments concerning our manuscript entitled " Analysis of lipid of green coffee by ultra-performance liquid chromatography-time of flight-tandem mass spectrometry" (molecules-1808702).
Reviewer 2:
The authors have made significant improvements to the manuscript based on the reviewers' comments and the manuscript is suitable for publication. However, there are minor flaws that need to be corrected before it is accepted for publication The authors can try to again rectify the overall grammatical errors throughout the manuscript. I have listed some of the major errors below.
- The terms 'green coffee' and 'green coffee oil' are used interchangeably throughout the text. Please keep it uniform
Response: It was revised. 'green coffee oil' was replaced with 'green coffee'.
- The author names are all capitalized in the manuscript and in references. Please use a uniform style and remove capitalization.
Response: It was revised. Please see the part of ‘name’ and ‘references’.
- Section 2.1.1. "Determination of fatty acid composition by gas chromatography" is a title and should not be referenced. Remove the references for the title. The same applies to Section 2.1.2.
Response: It was revised. The references for the title had be removed.
- Section 2.2 is missing and make the title font and italics uniform.
Response: It was revised. The title number has been corrected.
- Section 2.1.2. It is not clear what the authors mean by the statement "And the mass number range 120 for mass spectrometry was m/z 100-1200". Correct the grammatical errors throughout the manuscript.
Response: It was revised. This sentence was revised to " And The mass number collected by mass spectrometry ranges from m/z 100 to 1200".
- Section 3.2: Please edit the statement ' It could be seen the Figure 1(A)'. Just mention as 'Fig 1A represents....'
Response: It was revised. Please see the part of the Section 3.2, and marked in red.
- Fig 3 legend should be 'The composition of sterols, sphingosine.."
Response: It was revised.

Reviewer 3 Report
good work
Author Response
no

This manuscript is a resubmission of an earlier submission. The following is a list of the peer review reports and author responses from that submission.
Round 1
Reviewer 1 Report
Manuscript ID: molecules-1808702
Journal: Molecules
Article
Title: Analysis of lipid of green coffee by ultra-performance liquid
chromatography-time of flight-tandem mass spectrometry
Authors: LIU Yijun, CHEN Min *, LI Yimin, FEN Xingqin, CHEN Yulan, LIN Lijing*
This article by LIN Lijing et al. provides a detailed survey of the lipid content in green coffee by HPLC-TOF-MS/MS. A total of 214 lipids of different classes has been revealed. The most abundant fatty acids (FAs) have been identified and quantified, and the ratio of saturated, unsaturated and polyunsaturated FAs has been determined. The applied methodology appears robust, sensitive and accurate and the paper adds an interesting new piece of information in food analysis.
However, the presentation is scarce and a more accurate general editing and review of the whole manuscript is requested.
After that, the manuscript may be considered for publication in Molecules.
Minor comments:
Avoid the use of “etc.” all over the manuscript.
Lines 30-32: please avoid repetition of the same word in a sentence, when possible, and rephrase the period.
Line 38: control and uniformly apply the editorial rules for references.
Line 39: please rephrase the period: “researcher at home and abroad”.
Line 58: “…, the analysis of lipids in samples…”: please specify which kind of samples.
Line 82: Is “Determination of fatty acid composition by gas chromatography” a title of the paragraph? If not, uniform the font size and add a full stop at the end.
Lines 91-96. Add the correct punctuation and the verbs.
Line 99: same comment as for line 82.
Line 113: cancel “where”
Line 120: change “with an ESI source” with “coupled with an ESI source”
Line 120: check the sentence “…and the mass number range for mass ….:50.000”: it does not make any sense.
Line 134: “linoleic acid (C18:2n6c) 43.39%” has been repeated more time! Delete the repeated parts.
Line 142-144: rewrite the sentence “…which had more…and content.” in a clearer way.
Line 156: explain the acronym “ASG”
Line 157: delete the repeated part (Ceramide Esterified…sphingosine)
Line 169: change “parent ion” with “precursor ion”
Line 171-172: change was with displays and/or shows.
Line 172: change “and” with “where”
Line 179: as for line 169.
Line 179: change “.” with “,”
Line 183, 188, 189: as for line 169.
Line 218: change “0-3,” with “0-3.”
p. 7: Table 1 presents a break and no 30 jumps to no. 61
p. 8: Table 1 presents a break and no 104 jumps to no. 149
Lines 269-270: five components are listed, but only three amounts are reported.
Reviewer 2 Report
Review for molecules-1808702
Brief summary/ General comments
The manuscript aimed to characterise the types and diversity of lipids in green coffee via mass spectrometry, and identified various classes of lipids in green coffee. However, the findings of this study does not add considerable value to the literature of identified lipids in coffee, particularly as the green coffee origin/processing prior to the chemical extraction was not described. The authors did not identify the gap or motivation for the novelty of this study. It is also not clearly presented if the authors used an untargeted approach or a targeted approach to profile the lipids. If this was done using a untargeted approach, then the presentation of results in terms of % of lipids is misleading and inaccurate as this specific single HPLC or GCMS method may not fully capture the true profile of lipids. In addition, no information on molecular mass, mass accuracy, identification level or quantification of the lipids was provided.
Specific comments:
Abstract: The author should begin the abstract with the motivation or aim of the manuscript study, rather than beginning with the method/analysis.
Introduction line 70: the authors mentioned “for the basic development and utilisation of green coffee”, which can be elaborated further to demonstrate the motivation of this study.
Method line 73: “green coffee variety” description is arbitrary. There is no information on the origin and form of coffee (beans or ground coffee or coffee oil?). There is also no description on how the green coffee is processed before the extraction method (line 82). Such information is critical, considering the aim of the manuscript to characterise the lipids found in green coffee.
Method 2.1.2: if coffee powder was used in this study, natural variation throughout the coffee sample/origin is expected. Were replicates of the coffee performed for extraction and MS analysis? Furthermore, it is not clear if the authors used an untargeted approach or a targeted approach. Line 99 mentioned “MS/MS” but the method referenced seems to be a targeted approach?
Line 121: “100-1200: 50.00”- the value of 50.00 is not clear. Please elaborate.
Line 122: There is no mention of which lipid classes were captured by GCMS or HPLC-MS, and how quantification of the lipid classes were performed.
Line 132: an absolute value or 111.48mg/g is stated. Were replicates performed and if so, the standard deviation per sample replicate should be reported.
Line 133-136: it is misleading to present the findings of lipid classes as % as there is no evidence that all lipids/lipid classes were profiled through this specific HPLC method. In addition, the authors did not describe the cleavage patterns (line 155) used for the various lipid species or how they dealt with lipids that did not conform the used cleavage patterns in the text. This information should be provided as a supplementary material.
Line 151: were any MS or MS/MS databases used in the annotation of identification of the lipids?
Line 156: ASG is abbreviated. Please spell out at first mention.
Line 168-175: this paragraph details the parent ion, fragments and adducts of ASG 29:1 in both ionisation modes but is incredibly different to read. This information would be more suitable in a table form with several columns detailing the various molecular masses of fragments/adducts in each ionisation mode.
Line 197-199: this sentence can be misleading. The authors should explicitly state that this ‘profile of lipids’ were obtained using this single specific method. A different untargeted method would lead to a different “profile of lipids”.
Line 239-241: The quantification of each lipid class should be described and the list of commercial lipid standards used should be mentioned in the manuscript (in method section) for clarity and for knowledge on the identification level of the lipids- for example, which lipids’ identity were confirmed with an authentic standard and which lipids were tentatively identified.
Line 253-256, line 260: The authors describe the functions of dietary triglycerides and phospholipids. Do the authors anticipate that green coffee oil will be incorporated into our diet and if this can contribute to a considerable portion of dietary triglycerides or phospholipids intake, in comparison to coffee consumed as a beverage?
Line 257-258: in addition to being absorbed by the body, dietary phospholipids may also be act upon by the gut microbes to produce various phospholipids and choline.
Conclusion: the study highlighted the main findings of the study. However, the authors did not describe how this method was “fast, sensitive,… and reproducible” (line 270-271) in the earlier sections of the manuscript such as mass accuracy, day-to-day variation, quantification efficacy, etc.
Table 1: the identification level of lipids could be added into the table.
Reviewer 3 Report
The authors utilized HPLC-TOF-MS/MS analysis for analyzing lipid components present in a particular green coffee variety, Arabica coffee beans. The article needs significant improvement in structure and grammar before it is suitable for publication.
1. The authors should try to correct all grammatical and English language usage errors in the manuscript to ensure that the flow of the paper is clear and understandable. In several instances, the tenses (past and present) have been used interchangeably. I suggest the authors take the help of an English language editing service to work on the grammatical flaws in the manuscript.
2. There are several papers that have utilized LC-MS as a strategy to identify lipid components in coffee beans and the authors have not cited those papers. It is not clear what the novelty in their approach or results is. The authors do mention that Ref 17 and 18 have shown similar results previously. I recommend the authors do a thorough literature review of the subject and cite all relevant papers which utilize LC-MS for lipid composition analysis in coffee beans.
3. The data is not present in a clear and concise manner. For example, Table 1 lists all 181 different lipid species, but the authors have left gaps in between.